# New drug submissions in Canada and a comparison with the Food and Drug Administration and the European Medicines Agency: Cross-sectional analysis

Joel Lexchin [1,2,3] *

1 School of Health Policy and Management, York University, Toronto, Ontario, Canada, 2 Faculty of Medicine, University of Toronto, Toronto, Ontario, Canada, 3 University Health Network, Toronto, Ontario, Canada

* jlexchin@yorku.ca

## Abstract

### Background

Health Canada posts the outcomes of all New Drug Submissions. In some cases, companies have withdrawn submissions or submissions have been rejected by Health Canada for new active substances (NAS). This study explores the reasons for those decisions and compares them with decisions made by the Food and Drug Administration (FDA) and the European Medicines Agency (EMA).

### Methods

This is a cross-sectional analysis. Submissions for NAS between December 2015 and December 2022 were identified along with the original indications for the NAS, the information that Health Canada had available and the reasons for its decisions. Similar information was sourced from the FDA and the EMA. Their decisions were compared to those made by Health Canada. The time between decisions by Health Canada, the FDA and the EMA were calculated in months.

### Results

Health Canada considered 272 NAS and approved 257. Sponsors withdrew 14 submissions for 13 NAS and Health Canada rejected submissions for 2 NAS. The FDA approved 7 of these NAS and the EMA approved 6, rejected 2 and submissions were withdrawn by 2 companies. Health Canada and the FDA considered similar information in 4 of 7 cases. Indications were the same except in one case. The FDA made decisions a mean of 15.5 months (interquartile range 11.4, 68.2) before companies withdrew their submissions from Health Canada. There were 5 cases where Health Canada and the EMA considered the same information and in 2 of those the outcome was different. Health Canada and EMA decisions were generally made within 1–2 months of each other. Indications were the same in all cases.

**Data Availability Statement:** All relevant data are within the paper and its supporting information.

**Funding:** The author received no specific funding for this work.

**Competing interests:** In 2019-2022, Joel Lexchin received payments for writing a brief on the role of promotion in generating prescriptions for two legal firms. He is a member of the Foundation Board of Health Action International and the Board of Canadian Doctors for Medicare. He receives royalties from University of Toronto Press and James Lorimer & Co. Ltd. for books he has written.

## Conclusions

Differences in decision making by regulators are due to more than the data which with they are presented, the timing of the presentations and the indications for the drugs. Regulatory culture may have influenced decision making.

## Introduction

Before new active substances (NAS, new drugs that have never been marketed before in Canada in any form) can be approved, sponsors need to present Health Canada with evidence of their efficacy, safety and manufacturing quality [1]. (The equivalent of the NAS designation by the United States Food and Drug Administration (FDA) is new molecular entity.) Since December 2015, Health Canada has been posting the outcomes of New Drug Submissions on its Drug and Health Product Submissions Under Review website [2]. Outcomes include: canceled by sponsor; Notice of Compliance (market authorization); Notice of Compliance with conditions (market authorization with requirements for postmarket studies); Notice of Deficiency–Withdrawal (NOD-W, company has not adequately responded to Health Canada's decision that there is insufficient evidence to proceed with the review); and Notice of Non-Compliance–Withdrawal (NON-W, after the review has been completed, the company has not adequately responded to Health Canada's decision that the evidence does not warrant approval).

Health Canada, like other regulatory agencies, maintains that its decisions are based on an objective scientific analysis of the data that is presented to it by companies seeking approval for their new drugs [3]. If this characterization is correct, then the expectation is that well-resourced regulators from different high-income jurisdictions would reach similar decisions when considering the same data for the same indications. On-the-other hand, if other factors unrelated to the data package such as socio-cultural aspects come into play, then different regulators could reach contrasting decisions.

This study investigates how often submissions for NAS are either withdrawn by companies or rejected by Health Canada and the reasons for those decisions. It also compares Health Canada's decisions with those made by the FDA and the European Medicines Agency (EMA) for those same NAS and examines congruence and discrepancies in regulatory decision making.

## Methods

### Health Canada data

NAS where the sponsors withdrew the submission or they received either a NOD-W or a NON-W were identified through the New Drug Submissions on its Drug and Health Product Submissions Under Review website and the generic name of the NAS, the month and year when the submissions were concluded and their indication [2] were recorded on an Excel spreadsheet. Dates were only given by month and year and decisions were assumed to take place on the first day of each month. Hyperlinks from the Drug and Health Product Submissions Under Review website to the Drug and Health Product Register [4] were used to retrieve the reasons for Health Canada's decisions along with the key information that the companies provided for their drugs and this information was entered onto the same spreadsheet.

The total number of NAS approved by Health Canada during the study period (December 2015 to December 2022) was calculated from annual reports from Health Canada. Reports are available by emailing publications@hc-sc.gc.ca.

### FDA and EMA data

The websites of the FDA [5] and the EMA [6] were searched for the NAS in the cases where the Canadian sponsors withdrew their submissions or where Health Canada rejected the submission to determine the status of the NAS in those jurisdictions. If the NAS were listed on the websites the following information was recorded: the date when decisions were taken on the NAS, the key information that the companies provided the regulators for their drugs and the indication for the drugs. In the case of the FDA this information was in the Review document and it was in the European Public Assessment Report or the Withdrawal Assessment Report for the EMA.

### Data analysis

The median time in months was calculated was calculated between Health Canada and FDA/EMA decisions. Key information provided by the companies to the three regulators was categorized as "similar", "different" and "unable to determine" and compared. Decisions about the similarity of information were made based on the presence of information such as the number of pivotal studies, the study phase, their methodology (e.g., randomized, blinded, controlled), the type of population enrolled, the number of patients and other drugs that may have been administered during the study.

Calculations were done using Prism 9.5.1 (GraphPad Software, LLC).

### Ethics

Data were gathered by a single individual from February 19–22, 2023. All data were publicly available and ethics approval was not required. All data collected for this study are available in the S1 File. No patients were involved in this study.

## Results

Health Canada considered New Drug Submissions for 272 NAS from December 2015 until the end of 2022 and approved 257 of them. Fifteen NAS were not approved: sponsors withdrew 14 submissions for 13 NAS (a submission for ataluren was filed twice) and two submissions received a NON-W. One product, finerenone, was approved after a second submission. (There was no information about why the company withdrew the second submission for ataluren).

In 11 cases the company withdrew the submission after Health Canada identified deficiencies in the data that would have precluded approval. Health Canada generally provided few to no details about the data deficiencies that it identified (Table 1). In one case (lasmiditan), Health Canada did not identify any deficiencies but could not reach agreement with the sponsor on the interpretation of the cardiovascular data and resulting content in the Product Monograph (equivalent to the Label (FDA) and the Summary of Product Characteristics (EMA)). In another case (sirukumab) the reasons why the company withdrew the submission were not clearly stated. Health Canada rejected one submission (cinnarizine + dimenhydrinate) because the outcome was not considered valid and a second submission (volanesorsen) because of an unfavourable benefit to risk ratio (Table 1).

**Table 1. New active substance submissions withdrawn by company or rejected by Health Canada.**

| Generic name | Health Canada decision | Reason for decision |
|---|---|---|
| Aducanumab | New Drug Submission withdrawn by company | Health Canada concluded that clinical efficacy and safety data that were provided did not support the clinical benefit of using aducanumab for the proposed indication |
| Alvimopan | New Drug Submission withdrawn by company | Health Canada found deficiencies in the Chemistry & Manufacturing component of the New Drug Submission |
| Amisulpride | New Drug Submission withdrawn by company | Health Canada had identified deficiencies in the package that would have precluded issuing an approval |
| Ataluren* | New Drug Submission withdrawn by company | Health Canada had identified some deficiencies in the data that would have precluded issuing an approval |
| Cilostazol | New Drug Submission withdrawn by company | Health Canada identified some deficiencies in the package that would have precluded issuing an approval |
| Cinnarizine + dimenhydrinate | Rejected by Health Canada | Sponsor had not submitted sufficient supportive information to confirm that the Mean Vertigo Symptom was a valid Patient Reported Outcome which could be used in the present day regulatory environment to support authorization for the proposed indication |
| Emapalumab | New Drug Submission withdrawn by company | Health Canada had identified deficiencies in the clinical data that would have precluded issuing an approval |
| Finerenone† | New Drug Submission withdrawn by company | An initial review by Health Canada of the renal outcomes trial identified a number of uncertainties |
| Human heterologous liver cells | New Drug Submission withdrawn by company | Health Canada had identified some deficiencies in the data that would have precluded issuing an approval |
| Lasmiditan | New Drug Submission withdrawn by company | Health Canada did not identify any deficiencies in the data packages, however, Health Canada and the sponsor could not reach agreement on the interpretation of the cardiovascular data and resulting content in the Product Monograph‡ |
| Lorcaserin hydrochloride | New Drug Submission withdrawn by company | Health Canada had identified some open questions about the data that would have precluded issuing an approval and the company could not address the questions |
| Panobinostat | New Drug Submission withdrawn by company | Health Canada had identified a major deficiency in the information provided that precluded continuation of the review |
| Roxadustat | New Drug Submission withdrawn by company | Company response to a Notice of Non-Compliance left remaining unresolved issues |
| Sirukumab | New Drug Submission withdrawn by company | At the time of the cancellation, the clinical, quality and labelling reviews were pending |
| Volanesorsen | Rejected by Health Canada | Health Canada concluded that the risks and uncertainties, particularly the unpredictable events of severe thrombocytopenia, outweigh the evidence of benefit |

*New Drug Submission withdrawn twice by company, no information about reason for second withdrawal

†Approved by Health Canada on subsequent New Drug Submission

‡Equivalent to FDA Label and EMA Summary of Product Characteristics

## Health Canada and FDA comparison

Health Canada and the FDA both reviewed 7 of the 15 NAS, all of which the FDA approved (Table 2). Eight NAS were not found on the FDA website. Since the FDA does not report on submissions it rejects it is not clear if these NAS were submitted to the FDA and rejected or never submitted. The indications for the NAS were the same for both regulators except for amisulpride which was indicated for schizophrenia in Canada and the prevention of nausea and vomiting in the US (Table 3). Different outcomes between Health Canada and FDA occurred although similar information was submitted to the regulators in 4 cases. The information was different for amisulpride and it could not be determined if the information was similar or different for lasmiditan (S1 Table and Table 4). (There was no information about lorcaserin hydrochloride on the FDA website).

The FDA approved the 7 NAS a median of 15.5 months (interquartile range 11.4, 68.2) before companies withdrew their submissions from Health Canada's consideration (Table 2). On-the-other hand, companies withdrew their submissions from Health Canada 32.0 months

**Table 2. Date of decisions by Health Canada, FDA and EMA.**

| Generic name | Date of Health Canada decision | | Date of FDA decision | | Date of EMA decision | | |
| --- | --- | --- | --- | --- | --- | --- | --- |
| | Company withdrawal of submission | Rejection | Approval | Removal from market | Approval | Company withdrawal of submission | Rejection |
| Aducanumab | 2022-05-01 | | 2021-07-06 | | | 2022-04-20 | |
| Alvimopan | 2017-03-01 | | 2008-05-20 | | | | |
| Amisulpride | 2021-02-01 | | 2020-02-26 | | | | |
| Ataluren* | 2016-03-01 | | | | 2014-07-31 | | |
| Cilostazol | 2022-07-01 | | | | | | |
| Cinnarizine, dimenhydrinate | | 2018-11-01 | | | | | |
| Emapalumab | 2021-02-01 | | 2018-11-20 | | | | 2021-01-07 |
| Finerenone | 2021-10-01 | | | | 2022-02-16 | | |
| Human heterologous liver cells | 2018-07-01 | | | | | | 2015-12-21 |
| Lasmiditan | 2021-01-01 | | 2019-10-11 | | 2022-08-17 | | |
| Lorcaserin hydrochloride | 2018-02-01 | | 2012-06-27 | 2020-09-17 | | | |
| Panobinostat | 2016-06-01 | | 2015-02-23 | 2022-03-24 | 2015-08-28 | | |
| Roxadustat | 2021-10-01 | | | | 2021-08-18 | | |
| Sirukumab | 2017-10-01 | | | | | 2017-10-26 | |
| Volanesorsen | | 2018-11-01 | | | 2019-05-03 | | |

*New Drug Submission withdrawn twice by company, no information about reason for second withdrawal

(lorcaserin hydrochloride) and 70.7 months (panobinostat) before the drugs were removed by the FDA for safety reasons (Table 2).

## Health Canada and EMA comparison

Health Canada and the EMA jointly reviewed 10 NAS. The EMA approved 6 submissions, rejected 2 and submissions were withdrawn by 2 companies (Table 2). The indications for the 10 NAS were the same in Canada and in Europe (Table 3). One NAS was designated an orphan drug (lorcaserin hydrochloride) and was still under review, two NAS were not found on the EMA website (alvimopan and cinnarizine + dimenhydrinate) and two drugs (amisulpride and cilostazol) were nationally approved in Europe, i.e., only approved in selected countries. The EMA reports on the outcomes of all submissions it receives and therefore it is probable that there were no submissions for the two NAS not found on its website.

There were 5 cases where Health Canada and the EMA considered the same information and in 2 of those the outcome was the different–in Canada the company withdrew the submissions versus Europe where the NAS were approved. In the case of volanesorsen where the information was different the outcome was different–the drug was rejected by Health Canada versus approved by the EMA. In the other 4 cases it could not be determined if the information was the same or different (S1 Table and Table 5).

**Table 3. Indication for new active substances: Health Canada, FDA, EMA.**

| Generic name | Health Canada indication | FDA indication | EMA indication |
|---|---|---|---|
| Aducanumab | Disease modifying treatment for Alzheimer's disease in adults | Treatment of Alzheimer's disease | Disease modifying treatment in adult patients with Alzheimer's disease at the mild cognitive impairment (MCI) or mild dementia stage |
| Alvimopan | Accelerate the time to upper and lower gastrointestinal (GI) recovery following surgery | Accelerate the time to upper and lower gastrointestinal recovery following partial large or small bowel resection surgery | |
| Amisulpride | Treatment of acute and chronic schizophrenic disorders | Prevention of postoperative nausea and vomiting | |
| Ataluren | Treatment of patients with Duchenne Muscular Dystrophy | | Treatment of Duchenne muscular dystrophy |
| Cilostazol | Improvement of maximal and pain-free walking distance in patients with intermittent claudication | | |
| Cinnarizine + dimenhydrinate | Treatment of vertigo | | |
| Emapalumab | Treatment of pediatric and adult patients with primary hemophagocytic lymphohistiocytosis | Treatment of adult and pediatric (newborn and older) patients with primary hemophagocytic lymphohistiocytosis | Treatment of primary haemophagocytic lymphohistiocytosis |
| Finerenone | Delay progression of kidney disease and to reduce the risk of major adverse cardiovascular events | | Treatment of chronic kidney disease |
| Human heterologous liver cells | Treatment of pediatric patients from birth to less than 3 years of age suffering from severe urea cycle disorders | | Treatment of paediatric patients from birth to less than 6 years of age with urea cycle disorders |
| Lasmiditan | Acute treatment of migraine | Acute treatment of migraine | Acute treatment of the headache phase of migraine attacks |
| Lorcaserin hydrochloride | Adjunct to a reduced-calorie diet and increased physical activity for chronic weight management in adult patients | No information on FDA website about indication | |
| Panobinostat | Used in combination with bortezomib and dexamethasone, for the treatment of patients with multiple myeloma | In combination with bortezomib and dexamethasone...indicated for the treatment of patients with multiple myeloma | In combination with bortezomib and dexamethasone, is indicated for the treatment of patients with multiple myeloma |
| Roxadustat | Treatment of anemia due to chronic kidney disease | | Treatment of anaemia in adult patients with chronic kidney disease |
| Sirukumab | Treatment of moderately to severely active rheumatoid arthritis | | Treatment of moderately to severely active rheumatoid arthritis |
| Volanesorsen | Adjunct to diet for the treatment of patients with familial chylomicronemia syndrome | | Adjunct to diet for the treatment of patients with familial chylomicronemia syndrome |

**Table 4. Comparison of company supplied information and outcomes—Health Canada and the FDA.**

| Generic name | Information Assessed by Health Canada and the FDA | | | Regulatory outcome | |
|---|---|---|---|---|---|
| | Similar | Different | Unable to determine | Health Canada | FDA |
| Aducanumab | X | | | Company withdrew submission | Approved |
| Alvimopan | X | | | Company withdrew submission | Approved |
| Amisulpride | | X | | Company withdrew submission | Approved |
| Emapalumab | X | | | Company withdrew submission | Approved |
| Lasmiditan | | | X | Company withdrew submission | Approved |
| Lorcaserin hydrochloride | | | No information on FDA website about information assessed by FDA | Company withdrew submission | Approved |
| Panobinostat | X | | | Company withdrew submission | Approved |

Table 5. Comparison of company supplied information and outcomes–Health Canada and the EMA.

| Generic name | Information Assessed by Health Canada and the EMA | | | Regulatory outcome | |
| --- | --- | --- | --- | --- | --- |
| | Similar | Different | Unable to determine | Health Canada | EMA |
| Aducanumab | X | | | Company withdrew submission | Company withdrew submission |
| Ataluren* | X | | | Company withdrew submission | Approved |
| Emapalumab | X | | | Company withdrew submission | Rejected |
| Finerenone | | | X | Company withdrew submission | Approved |
| Human heterologous liver cells | X | | | Company withdrew submission | Rejected |
| Lasmiditan | | | X | Company withdrew submission | Approved |
| Panobinostat | X | | | Company withdrew submission | Approved |
| Roxadustat | | | X | Company withdrew submission | Approved |
| Sirukumab | | | X | Company withdrew submission | Company withdrew submission |
| Volanesorsen | | X | | Rejected | Approved |

*New Drug Submission withdrawn twice by company, no information about reason for second withdrawal

The EMA approved the 6 NAS a median of 1.6 months (interquartile range -11.8, 9.5) months after the submissions were either withdrawn in Canada or Health Canada rejected the submissions. In 3 cases (aducanumab, emapalumab and sirukumab) the EMA rejection or the company withdrawal occurred within 1 month of the submission withdrawal in Canada. In the fourth case (human heterologous liver cells) the EMA rejection was 30.8 months before the company withdrew the submission in Canada (Table 2).

## Discussion

Over a 7-year period, Health Canada approved 257 NAS, rejected 2 submissions and companies withdrew 14 submissions for 13 NAS. Nearly all the submissions to Health Canada were withdrawn because the agency identified data deficiencies although these deficiencies were not generally further specified. Despite all 15 drugs not reaching the Canadian market (one was approved on a subsequent submission), 7 were approved by the FDA and 6 were approved by the EMA. In one case (amisulpride), the indications were different and that could have been the reason for the different decisions by Health Canada and the FDA.

Time differences and differences in the content of the information provided to the regulators are unlikely to be the sole explanations for different regulatory outcomes. Although the FDA approved drugs a median of 15.5 months before submissions were made to Health Canada, the latter identified deficiencies in two submissions that led companies to withdraw them 32 and 71 months before the FDA withdrew the products. Decisions by the EMA were mostly contemporaneous with those by Health Canada. In 4 cases where Health Canada and the FDA had the same information the outcome was different and this also happened in 2 cases with the EMA.

Other studies have explored differences in decisions between regulatory agencies, usually the FDA and the EMA. The FDA gave a priority review to 12 tyrosine kinase inhibitors whereas the equivalent EMA designation was not used for any of them. Conversely, the FDA granted Accelerated Approval for 6 of them and the EMA used the equivalent conditional approval for 4 [7]. Most of the 21 drug/indications for oncology drugs approved by both the FDA and the EMA between 2009 and 2013 relied on identical pivotal trials. However the two agencies often used different regulatory pathways; 57% of the indications received either FDA Accelerated Approval or EMA Conditional Marketing Authorization, and regular approval by

the other agency [8]. In another analysis of 42 oncology drugs approved for 100 indications by the EMA between 1995 and 2008, there was a discrepancy between the EMA and the FDA in 47 of these indications [9]. Five FDA-approved drugs between 2017–2020 were refused marketing authorization by the EMA due to unfavourable, benefit-to-risk assessments [10]. Conversely, the FDA initially rejected submissions for safety reasons for 12 of 37 drugs with novel mechanisms of action approved first in Europe and/or Canada [11].

The variation in decision making may reflect distinct therapeutic cultures in the different jurisdictions. These "therapeutic cultures arise from networks of actors that produce regulatory policy, determine testing standards, and ultimately decide on market access for new drugs" [12]. The assumption that regulators do not differ across different jurisdictions would be a mistake. As Daemmrich points out, drug regulation is the outcome of the intersection between values, science, medical culture, patient needs and expectations and politics [12]. The differences in the regulation of nonsteroidal anti-inflammatory drugs in the US and the United Kingdom documented by Abraham and Davis are an illustration of how regulation has differentially evolved [13]. Variations in how regulators act can also be seen in the different ways that they compose their advisory committees, how they structure their interactions with industry, and the extent to which they integrate patients into their processes.[14–16]

Two comparisons of how the FDA and the EMA make decisions on oncology drugs found that they manage uncertainty differently. The conclusion from one study was that the FDA was "more open to take risks and base approval on less robust data in order to guarantee quicker access to anticancer medications" [17]. The second study did not find any data showing that the FDA took more risks but did conclude that the two agencies approached risk differently [8]. Both studies illustrate that informal factors, while secondary to the data in driving decisions, play an important role in the drug regulation process. When it comes to how it handles drug safety issues, Health Canada has been characterized as a "shadow regulator", i.e., one that shadows decisions made by other regulators that have a reputation for expertise [18]. If that characterization carries over to drug approvals it may play a role in explaining the differences between Health Canada, the FDA and the EMA.

## Limitations

Even when the regulators examined the same information, they may have interpreted that information differently. This possibility was not explored in this study. Differences in the prevalence of the disease in the different jurisdictions, the perceived need for the treatment and the presence of other treatments are not taken into consideration in regulatory decisions and therefore should not have accounted for different decisions. Data were acquired by a single individual and that might have introduced biases.

## Conclusion

Differences in decision making by regulators are due to more than just the data which with they are presented, the timing of that presentation and the indications for the drugs. Recognizing that differences are also due to therapeutic culture needs to be taken into consideration when there are calls for Canada to automatically accept regulatory decisions taken in other jurisdictions.

## Supporting information

**S1 Checklist. STROBE statement—checklist of items that should be included in reports of observational studies.**
(DOCX)

**S1 File. Information from Health Canada, Food and Drug Administration and European Medicines Agency used in the analysis of the drugs considered in this study.**
(XLSX)

**S1 Table. Summary of pivotal information considered by Health Canada, FDA and EMA in decision-making.**
(DOCX)

## Author Contributions

**Conceptualization:** Joel Lexchin.

**Data curation:** Joel Lexchin.

**Formal analysis:** Joel Lexchin.

**Investigation:** Joel Lexchin.

**Methodology:** Joel Lexchin.

**Writing – original draft:** Joel Lexchin.

**Writing – review & editing:** Joel Lexchin.

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
