## [Decision Letter · Decision Letter 0]

3 May 2023

PONE-D-23-06043New drug submissions in Canada and a comparison with the Food and Drug Administration and the European Medicines Agency: cross-sectional analysis

PLOS ONE

Dear Dr. Lexchin,

Thank you for submitting your manuscript to PLOS ONE. After careful consideration, we feel that it has merit but does not fully meet PLOS ONE’s publication criteria as it currently stands. Therefore, we invite you to submit a revised version of the manuscript that addresses the points raised during the review process.

We look forward to receiving your revised manuscript.

Kind regards,

Hideki Maeda, Ph.D.

Academic Editor

PLOS ONE

Journal Requirements:

- https://doi.org/10.1186/s40545-021-00375-y

- https://doi.org/10.1177/0020731420979824

- https://doi.org/10.2190/HS.42.1k

In your revision ensure you cite all your sources (including your own works), and quote or rephrase any duplicated text outside the methods section. Further consideration is dependent on these concerns being addressed

"Unfunded study"

Additional Editor Comments (if provided):

We have completed our review of the manuscript. Your manuscript is well written and worth publishing. However, some corrections are needed. Please read carefully the sections the reviewer points out and make the revisions.

Reviewers' comments:

Reviewer's Responses to Questions

**Comments to the Author**

1. Is the manuscript technically sound, and do the data support the conclusions?

Reviewer #1: Yes

Reviewer #2: Yes

2. Has the statistical analysis been performed appropriately and rigorously? 

Reviewer #1: N/A

Reviewer #2: Yes

3. Have the authors made all data underlying the findings in their manuscript fully available?

Reviewer #1: Yes

Reviewer #2: Yes

4. Is the manuscript presented in an intelligible fashion and written in standard English?

Reviewer #1: Yes

Reviewer #2: Yes

5. Review Comments to the Author

Reviewer #1: The manuscript would benefit to have the regulatory abbreviations in a table or as list sorted by agencies.

The conclusions in the abstract do not reflect the conclusion in the main manuscript: "for Canada to automatically accept regulatory decisions"

Reviewer #2: Thank you for giving me the opportunity to review the manuscript entitled "New drug submissions in Canada and a comparison with the Food and Drug Administration and the European Medicines Agency: cross-sectional analysis". This reviewer applauds the author's aim to evaluate the withdrawl and rejection of new indications by health canada. Withdrawls and rejections of new drug indications have become an important concern around the globe. Differences in the perception and submission of clinical trial evidence between regulatory agencies may result in growing disparities in the access to new medicines across countries. Therefore, this article is of high interest to physicians, regulators, and patients. Please see my detailed comments below:

Abstract:

- The author should specify if NAS include only original indications or if NAS also include supplemental indication approvals.

- The author should probably start by giving the total sample size of all NAS that were analyzed and then zoom-in on the withdrawls and rejections.

- A more nuanced conclusion is necessary.

Introduction:

- The author should provide a better introduction that explains why this study is relevant and unique. So far, only the technical regulatory terms are introduced without giving the reader more context why they should continue to read the manuscript.

Methods:

- "Key information provided by the companies to the three regulators was categorized as “similar”, “different” and “unable to determine” and compared." -> how exactly did the author decide upon these categories? This description is not very objective.

Results:

- Well done!

Discussion:

- Very good discussion of the results and comparison to previous studies.

- There are other reasons that may explain differences in regulatory decisions on new drug indications. First, the current standard of care may differ between countries. Second, there are difference in the way new indications are prescribed, reimbursed, and priced across nations (which could ultimately also influence the perception of regultators on the marketing authorization decisions). See https://doi.org/10.1007/s40258-022-00737-w and https://doi.org/10.1007/s10637-022-01227-5 for these points.

6. PLOS authors have the option to publish the peer review history of their article (what does this mean?). If published, this will include your full peer review and any attached files.

Reviewer #1: No

Reviewer #2: **Yes: **Daniel Tobias Michaeli

---

## [Author Response · Author response to Decision Letter 0]

4 May 2023

Reviewer #1: The manuscript would benefit to have the regulatory abbreviations in a table or as list sorted by agencies. 

My understanding is that PLoS ONE does not use a table of abbreviations but if the editors would like one, I would be happy to comply. 

The conclusions in the abstract do not reflect the conclusion in the main manuscript: "for Canada to automatically accept regulatory decisions".

The following sentence was added to the Conclusion in the Abstract: “Regulatory culture may have influenced decision making.”

Reviewer #2: Thank you for giving me the opportunity to review the manuscript entitled "New drug submissions in Canada and a comparison with the Food and Drug Administration and the European Medicines Agency: cross-sectional analysis". This reviewer applauds the author's aim to evaluate the withdrawal and rejection of new indications by health canada. Withdrawals and rejections of new drug indications have become an important concern around the globe. Differences in the perception and submission of clinical trial evidence between regulatory agencies may result in growing disparities in the access to new medicines across countries. Therefore, this article is of high interest to physicians, regulators, and patients. Please see my detailed comments below: 

I appreciate the supportive comment from the reviewer.

Abstract: 

The author should specify if NAS include only original indications or if NAS also include supplemental indication approvals. 

The Methods section of the Abstract now says “Submissions for NAS between December 2015 and December 2022 were identified along with the original indications for the NAS…”

The author should probably start by giving the total sample size of all NAS that were analyzed and then zoom-in on the withdrawals and rejections. 

The first sentence in the Results section of the Abstract now reads “Health Canada considered 272 NAS and approved 257.” 

A more nuanced conclusion is necessary. 

I am not clear how the reviewer would like the Conclusion to be worded. If he could be more specific, I would be happy to try and comply.

Introduction: 

The author should provide a better introduction that explains why this study is relevant and unique. So far, only the technical regulatory terms are introduced without giving the reader more context why they should continue to read the manuscript. 

I appreciate the reviewer pointing out the need for this type of statement. A penultimate paragraph in the Introduction has been added: “Health Canada, like other regulatory agencies, maintains that its decisions are based on an objective scientific analysis of the data that is presented to it by companies seeking approval for their new drugs. If this characterization is correct, then the expectation is that well-resourced regulators from different high-income jurisdictions would reach similar decisions when considering the same data for the same indications. On-the-other hand, if other factors unrelated to the data package such as socio-cultural aspects come into play, then different regulators could reach contrasting decisions.”

Methods: 

"Key information provided by the companies to the three regulators was categorized as “similar”, “different” and “unable to determine” and compared." -> how exactly did the author decide upon these categories? This description is not very objective. 

Under the Data Analysis subheading the following has been added: “Decisions about the similarity of information were made based on the presence of information such as the number of pivotal studies, the study phase, their methodology (e.g., randomized, blinded, controlled), the type of population enrolled, the number of patients and other drugs that may have been administered during the study.”

Results: 

Well done! 

Thank you.

Discussion: 

Very good discussion of the results and comparison to previous studies.

Thank you.

There are other reasons that may explain differences in regulatory decisions on new drug indications. First, the current standard of care may differ between countries. Second, there are difference in the way new indications are prescribed, reimbursed, and priced across nations (which could ultimately also influence the perception of regulators on the marketing authorization decisions). See https://doi.org/10.1007/s40258-022-00737-w and https://doi.org/10.1007/s10637-022-01227-5 for these points.

The reviewer is correct that the current standard of care may differ between countries. Those differences may in turn influence the type of indication(s) that drug companies seek approval for and therefore the data that they submit to regulatory authorities. However, to my knowledge, regulators only consider the quality of the data (e.g., do the trials demonstrate efficacy and safety) in making their decisions. The existence of other treatments for the same condition and the standard of care come into play when decisions are made about whether to include drugs on a formulary, whether it should be publicly funded, whether its use should be restricted to certain locations (e.g., hospitals) and whether prescribers need special training in order to be able to prescribe it.

I appreciate the reviewer providing me with the two studies. They reinforce his point that indication development is prioritized according to clinical value and disease prevalence and that these may vary across indications leading to different health technology assessment decisions. It is also possible that regulators may consider these differences in their decisions about whether or not to approve a new drug, but as far as I am aware there is no literature to support this speculation. I think that exploring this possibility would be a very interesting piece of research, but I feel that starting this discussion is outside the bounds of my study.

---

## [Decision Letter · Decision Letter 1]

24 May 2023

New drug submissions in Canada and a comparison with the Food and Drug Administration and the European Medicines Agency: cross-sectional analysis

PONE-D-23-06043R1

Dear Dr. Lexchin,

We’re pleased to inform you that your manuscript has been judged scientifically suitable for publication and will be formally accepted for publication once it meets all outstanding technical requirements.

Kind regards,

Hideki Maeda, Ph.D.

Academic Editor

PLOS ONE

Additional Editor Comments (optional):

Thank you for submitting your manuscript to PLOS ONE. I checked author's revisions and I also assess this study using the STROBE gideline. After careful consideration, I feel that it has merit for PLOS ONE’s publication.

Reviewers' comments:

Reviewer's Responses to Questions

**Comments to the Author**

1. If the authors have adequately addressed your comments raised in a previous round of review and you feel that this manuscript is now acceptable for publication, you may indicate that here to bypass the “Comments to the Author” section, enter your conflict of interest statement in the “Confidential to Editor” section, and submit your "Accept" recommendation.

Reviewer #2: All comments have been addressed

2. Is the manuscript technically sound, and do the data support the conclusions?

Reviewer #2: Yes

3. Has the statistical analysis been performed appropriately and rigorously? 

Reviewer #2: Yes

4. Have the authors made all data underlying the findings in their manuscript fully available?

Reviewer #2: Yes

5. Is the manuscript presented in an intelligible fashion and written in standard English?

Reviewer #2: Yes

6. Review Comments to the Author

Reviewer #2: (No Response)

7. PLOS authors have the option to publish the peer review history of their article (what does this mean?). If published, this will include your full peer review and any attached files.

Reviewer #2: **Yes: **Daniel Tobias Michaeli

---

## [Editor Report · Acceptance letter]

6 Jun 2023

PONE-D-23-06043R1 

New drug submissions in Canada and a comparison with the Food and Drug Administration and the European Medicines Agency: cross-sectional analysis 

Dear Dr. Lexchin:

I'm pleased to inform you that your manuscript has been deemed suitable for publication in PLOS ONE. Congratulations! Your manuscript is now with our production department. 

Kind regards, 

on behalf of

Professor Hideki Maeda 

Academic Editor

PLOS ONE